# Fish Protein Hydrolysate as Protein Enrichment in Texture-Modified Salmon Products

**DOI:** 10.3390/foods14020162

**Published:** 2025-01-08

**Authors:** Leena Prabhu, Aase Vorre Skuland, Paula Varela, Jan Thomas Rosnes

**Affiliations:** Nofima AS, Richard Johnsensgate 4, 4068 Stavanger, Norway; aase.vorre.skuland@nofima.no (A.V.S.); paula.varela.tomasco@nofima.no (P.V.); thomas.rosnes@nofima.no (J.T.R.)

**Keywords:** salmon, texture modified, IDDSI, protein enriched, fish protein hydrolysate, upcycled food

## Abstract

The aim of this study was to develop a chilled, texture-modified salmon product for dysphagia patients, enriched with dairy and fish hydrolysate proteins. The challenge was to create a product with appealing sensory qualities and texture that meets level 5 (minced & moist) of the IDDSI framework. Atlantic salmon (*Salmo salar*) was heat-treated (95 °C/15 min), blended, and reconstructed by adding texture modifiers, casein and whey protein, and enzymatically derived fish hydrolysate. The products were packaged in oxygen-free plastic trays, heat-treated to a core temperature of 95 °C for 15 min, chilled and stored at 4 °C for 29 days and analyzed for microbiology, instrumental texture, and sensory properties. The texture analyses showed that products with fish protein hydrolysate were softer than those only with casein and whey protein, a result also confirmed by the IDDSI fork pressure test. Quantitative descriptive analysis of salmon products revealed significant differences (*p* < 0.05) in sensory attributes within flavour (fish flavour), and texture (softness and adhesiveness) but there was no significant change in bitterness. The shelf-life study at 4 °C showed good microbiological quality of the product, and safety after 29 days with appealing sensory and textural properties, i.e., a product at IDDSI level 5 for age care facilities and commercial production was obtained.

## 1. Introduction

Dysphagia, or chewing and swallowing problems, is a common morbidity experienced by people who have suffered from stroke, Alzheimer’s, different surgeries on the head or neck, cancer and for older adults [1,2]. Depending on participant selection, screening or assessment tools, the prevalence of oropharyngeal dysphagia varies between 2.3% and 16%. It increases with the ageing of the population, going up to 26.7% for people older than 76 years of age [3]. More than 30% of hospitalized elderly patients with dysphagia suffer from malnutrition, which implies a lower functional capacity [4,5]. Loss of swallowing function can result in serious health issues, including dehydration, malnutrition, pneumonia, and decreased quality of life [6]. As people age, they are more likely to develop chronic diseases such as diabetes, cancer, and heart disease, as well as have an increased risk of frailty, cognitive decline, and disabilities. Improved nutrition is known to provide significant benefits for older adults. Many age-related diseases and conditions can be prevented or alleviated through tailored nutritional approaches [7]. One way to address this issue is by incorporating texture-modified products, which are designed to be easier to chew and swallow. These foods should also provide good nutritional value and be ready to eat, similar to regular food [8]. The texture can be adjusted based on the severity of the swallowing disorder, as outlined in guidelines that classify texture-modified foods, ranging from viscous soups to soft, solid products [9]. The International Dysphagia Diet Initiative (IDDSI) was developed during the last decade as a framework to define texture-modified foods and drinks, and this framework is being used in an increasing number of countries [10,11]. Texture modification helps individuals with dysphagia to absorb nutrients, thereby improving their nutritional status better. There are three main strategies for creating texture-modified foods suitable for these individuals. The first strategy involves providing a purée diet, made through mechanical blending. The second strategy requires thickening liquids using chemical agents, such as texturizing polymers. However, both strategies have drawbacks, including the potential deficiency of certain nutrients [12]. The third strategy, which is the most complex and least studied, involves the textural adaptation of intricate dishes [13,14]. Since texture modified products are ground up to a homogeneous mass, proteins are usually added to compensate for the reduced concentration due to dilution. The dairy proteins whey and casein are used for their functional properties. In recent years, there has been a great focus on sustainability in food production, and one strategy is to use hydrolysed proteins from by-products in the fishing industry. The fish processing industry generates significant amounts of protein-rich by-products that are often discarded or underutilized each year [15,16]. To upcycle proteins more effectively and sustainably for human health and consumption, several biotechnologies have been developed [17]. One such technology is the enzymatic hydrolysis of native proteins, which produces protein hydrolysates. Protein hydrolysates are defined as proteins that have been chemically or enzymatically broken down into peptides of varying sizes [18]. Fish protein hydrolysates (FPH) serve as excellent sources of amino acids and contain small fragments of biologically active peptides, which are considered beneficial for promoting health in relation to certain diseases [15,19,20]. Enzymatic hydrolysates often have a strong and sometimes bitter taste of the raw material from which they are extracted [20]. Therefore, when fish hydrolysates are used for new fish products, there is a challenge in finding the correct balance with other added ingredients and proteins.

Good nutritional statuses in older adults and those with different levels of dysphagia can keep them functioning longer, which has a bearing on their well-being and the desire for life. Many countries are facing an increased number of elderly people, and many must live at home for longer periods, outside nursing homes and hospitals. In many European countries, few commercially defined texture-modified products are available in the grocery store chill chain or for controlled distribution to domestic residents [5]. As such, there is a need to develop chilled, texture-modified products enriched in protein, ideally a “swallow-safe” bolus that is moist, cohesive, and slippery, and which is in accordance with IDDSI standards. Therefore, this work aimed to examine different recipes for texture-modifying a salmon product while utilizing sustainable proteins that come from residual raw material from fish.

## 2. Materials and Methods

### 2.1. Raw Materials

Farmed, skinless and boneless salmon (*Salmo salar*) belly loins (SALMA, Salmon brand, Bremnes, Norway) were packed in portions of 1000 ± 2 g, in sous-vide bags (PA/PE 70my, LietPak, Vilnius, Lithuania) and sealed using a 99.9% vacuum (Webomatic^®^, Supermax C, Bochum, Germany). The packages were frozen at −18 °C until further use, which would occur within 4 weeks.

### 2.2. Protein Enrichment

The additional proteins used were sodium caseinate (KAPA^TM^, JPR 1002, Armor proteins, Maen-Roch, France), salmon protein hydrolysate, (Hofseth BioCare AS, Midsund Norway), and whey protein concentrate (WPC 80, Tine SA, Nærbø, Norway).

### 2.3. Preparation of Samples

Frozen and vacuum-packed salmon were thawed at 4 °C for 20 ± 2 h in a container with cold water. The thawed fish packages were then cooked at 100 °C with 100% steam to a core temperature of 95 °C for 15 min in a preheated convection oven (MSCC61, Metos, Kerava, Finland). The temperature was monitored by three temperature probes (E-Val flex, Ellab, Hillerød, Denmark) inserted into the thickest part of three random fillets. After cooking, the packed fish fillets were cooled rapidly to 30 ± 5 °C in an ice slurry. Cooked fish, including the cook loss in the packages, was blended with sequentially added ingredients (Table 1) in a Thermomix^®^ (TM5, Vorwerk, Cloyes-sur-le-Loir, France) at a specific speed and time interval, as listed in Table 2. Two samples, sample S (enrichment: WPC 80 + sodium caseinate) and sample SFP (enrichment: WPC 80 + sodium caseinate + fish protein hydrolysate) were prepared. The homogenous mixture was stored at 4 °C until portioning and packaging.

The homogenized minced salmon was portioned (120 ± 2 g, 25 ± 2 mm height) to about 50% of a 280 mL volume in a food-grade HDPE tray with dimensions of 93 × 93 × 53 mm (562 RPC Bebo Food Packaging, Kristiansand, Norway, O_2_ transmission rate < 3.3 cm^3^/m^2^/24 h/atm, CO_2_ transmission rate 14.0 cm^3^/m^2^/24 h/atm at 23 °C, 0% RH). The samples were flushed with 100% nitrogen gas to replace O_2_ and simulate vacuum packaging to delay oxidative rancidity and inhibit the growth of aerobic microorganisms. Trays were sealed with Dynoseal ST 1575, a 90 μm sealing film (RPC Bebo Food Packaging Norway, O_2_ transmission rate < 35 cm^3^/m^2^/24 h/atm, CO_2_ transmission rate < 130 cm^3^/m^2^/24 h/atm at 23 °C, 0% RH) using a Dynopack VGA 462 sealing machine (RPC Bebo Food Packaging, Kristiansand, Norway).

A Steriflow Shaka^®^ autoclave (Roanne, France) was used in a static mode to pasteurize the fish samples at a core temperature of 95 °C for 15 min. The temperature was measured with temperature probes (E-Val flex, Ellab, Hillerød, Denmark). The trays were cooled immediately after autoclaving in an ice slurry bath for approximately one hour.

After cooling, all the trays except those for sensory analysis were packed in 250 × 350 mm PA/PE 90 μm bags (AZ-Pack, Kaunas, Lithuania). Trays for sensory analysis were packed in a 460 × 700 mm PA/PE 80 μm bags (AZ-Pack, UAB, Lithuania). To reduce the O_2_ content during storage at 4 °C, the bags were flushed with 100% N_2_ gas and sealed using CVP A-600 MAP packaging machine (CVP systems, Illinois, USA). Residual gas for both bags and trays were measured during storage using a CheckMate 3, O_2_/CO_2_ gas analyser (Dansensor, Ringsted, Denmark). To compare samples with varying storage durations, products for sensory analysis were produced on three dates: 29, 22, and 8 days before the evaluation. The samples from the first production date underwent texture, IDDSI fork test, colour, and microbiological analyses during the shelf-life testing.

### 2.4. Sensory Analysis

Quantitative Descriptive Analysis (QDA^®^) was performed in accordance with ISO standard 13299:2016 [21] by a highly trained sensory panel, following ISO standard 8589:2007 [22]. The analysis was conducted by nine assessors on duplicate evaluation of samples stored at 8, 22, and 29 days at 4 °C (*n* = 18). The panellists were trained according to ISO 8586:2012 [23]. A pre-trial session was conducted to develop a relevant vocabulary based on consensus of the panel, consisting of 7 odour attributes (sour, sweet, metallic, dairy, spice, fish and cloying), 3 appearance attributes (uniformity, dotted and glossy), 11 flavour attributes (sour, sweet, salt, bitter, umami, metallic, dairy, spice, fish, cloying and aftertaste) and 6 texture attributes (soft, fat, grainy, cohesive, adhesive and astringent). The full description of sensory attributes is provided in Appendix A.

Shortly before the sensory evaluation, samples were heated at 100 °C (moist heat mode) in the sealed HDPE trays to a core temperature of 60 ± 2 °C using a preheated convection oven (Electrolux air-o-steam, 260462, Vallenoncello, Italy). Immediately after heating, the samples were removed from the trays, upside-down, and cut into 6 equally sized squared portions (20 ± 1 g). Each portion was placed into a preheated (60 ± 2 °C) porcelain bowl labelled with a three-digit code, covered with an aluminum lid, and served to the assessors randomly in triplicates over four sessions with a short break midway. The eating temperature during evaluation was 50 ± 5 °C, considering the time taken to assess the samples. The sensory attributes were evaluated, with an intensity scale of 1 to 9 [24] where 1 = no intensity and 9 = high intensity. A computerized system EyeQuestion Software version 4.10.4 (Logic8 BV, Elst, The Netherlands) was used for code generation and data recording.

### 2.5. Colour Analysis

Colorimetric analysis was performed on samples (*n* = 3) on day 8, 15, 22, 29, 36, and 43 after production. A digital colour imaging system (DigiEye™, VeriVide Ltd., Leicester, UK) was used to record values in the CIE Lab colour space. The samples stored at 4 °C were removed from the tray and placed upside down in an illumination cube (standard daylight, 6400K). Surface images were taken with a calibrated digital camera (Nikon D80, 35 mm lens, Tokyo, Japan). DigiPix software (Version 2.8, VeriVide Ltd., Leicester, UK) was used to calculate L*a*b* values from the image, where L* indicates lightness of the sample and ranges from 0 (black) to 100 (white). While a* changes from −a (green) to +a (red) and b* values change from −b (blue) to +b (yellow). C* (Chroma/saturation) and h° (hue) were calculated using the following formulas: C* = (a^2^ + b^2^)^1/2^ and h* = arctan (b*/a*), respectively.

### 2.6. Texture Analysis

The firmness of samples was analyzed with a TA. XT Plus Texture Analyzer (Stable Micro Systems Ltd., Surrey, UK) equipped with a 5 kg load cell. Each sample was analyzed in triplicate with four repetitions on each sample (*n* = 3 × 4) on day 1, 8, 15, 22, 29, 36, and 43 after production. The samples were heated to a core temperature of 60 ± 2 °C and placed in a heating block (AccuBlock™, Labnet International, Edison Township, NJ, USA) to stabilize the temperature during measurement. The Firmness was expressed as the maximum peak force (Fmax) in Newtons (N) required for the 0.5 R cylinder probe to penetrate the samples to 12.5 mm, with a test speed of 1.50 mm/sec and trigger force of 5 g. The Exponent Software (Version 6.1.13.0.) (Stable Micro Systems Ltd., Surrey, UK) was used to analyze the data.

### 2.7. IDDSI Fork Pressure Test

To determine the appropriate IDDSI method, a preliminary test was conducted using the spoon tilt and fork pressure tests. The fork pressure test was selected because the samples were found to be too firm and sticky for the spoon tilt test. The IDDSI levels for the samples in this study ranged from 5 to 7.

The fork pressure test [25] was performed on the same samples, S and SFP, used for texture measurement. The test was performed in triplicate on day 1, 8, 22, 29, 36, and 43. Upon completion of the texture measurement, the samples were transferred directly from the heating block (60 ± 2 °C) and turned upside-down onto a porcelain plate to expose the smooth surface. A stainless-steel fork with a prong width of 1.5 cm was used to apply pressure. The thumb was placed onto the bowl of the fork just below the prongs, and then onto the sample, applying pressure until the thumb nail being blanched to white was observed [10]. Considering the time needed for each IDDSI fork pressure test, it is reasonable to assume that the sample temperature was 50 ± 5 °C during measurement.

### 2.8. Microbiological Analysis

A total of 25 ± 2 g of the sample was placed into a sterile stomacher bag (Separator 400, Grade products Ltd., Leicestershire, UK), diluted 10-fold with peptone salt water (0.1% *w*/*v* peptone and 0.85% *w*/*v* NaCl) and homogenized (SMASHER^®^, BioMérieux, Marcy-l’Étoile, France) for 2 min. Homogenates were serially diluted and 100 μL of each dilution was spread onto Plate Count Agar (PCA) (Merck KGaA, Darmstadt, Germany) and incubated at 37 °C for 72 h. Aerobic and anaerobic spore-forming bacteria were enumerated by spreading 100 μL of sample, heat-treated at 80 °C for 10 min, onto blood agar plates (CM0271, Oxoid, Hampshire, UK), and incubated according to NMKL 189.

### 2.9. Statistical Analysis

Statistical analysis was performed using Minitab^®^ 19 (Minitab, Coventry, UK). All data, except for the sensory analysis data, were analyzed using a general linear model (GLM), with a significance threshold set at *p* < 0.05 to identify significant differences between samples and storage time. A one-way analysis of variance was performed on the sensory data to assess if significant differences existed among the attributes between the samples. Tukey’s pairwise comparisons were also executed at a significance level of *p* < 0.05. All results are presented as mean ± standard deviation (SD) unless otherwise specified.

## 3. Results

### 3.1. Sensory Analysis

The average sensory scores on a selection of attributes from QDA^®^ (n = 18, mean ± S.D.) of sample S (without fish protein enrichment) and sample SFP (with fish protein enrichment) on storage day 8, 22, and 29 are shown in Figure 1. Only a few sensory properties (fish flavour, softness, and adhesiveness) showed significant difference.

The use of enrichment did not give any significant difference either due to the types of added protein or storage time in terms of dairy odour and flavour, fish odour, or bitter taste. There was a significant increase (*p* = 0.006) in fish flavour for the SFP samples between storage day 8 and 22. For softness, samples enriched with fish protein (SFP) were significantly softer (*p* = 0.012) on day 8 than samples without enrichment (S). Storage time gave significantly increased adhesiveness in sample S between storage day 22 to 29 (*p* = 0.340) according to Fisher Pairwise Comparisons.

Protein hydrolysates derived from animal by-products have, in earlier research, been reported to give an undesirable bitter taste [26,27,28,29]. Protein hydrolysates from salmon, mackerel, and herring heads and backbones were evaluated for their sensory properties by a highly trained sensory panel [30]. The results indicated that the hydrolysates made from herring had the most intense flavour, while those made from salmon were considered more palatable. In our study, no significant differences in bitterness were observed. It can be assumed that the level of enrichment used did not contribute to any bitter taste, or that the milk ingredients, such as whipping cream, sodium caseinate, and whey protein, helped to mask any potential bitterness.

The more adhesive a product is, the more cohesive it can be with moisture from added liquid and sodium caseinate. As reviewed by [31], the soluble form of caseins, such as sodium caseinate, tends to make certain foods sticky or ‘doughy’ by binding excessive amounts of water, resulting in a more adhesive and cohesive texture. In this study, the S and SFP samples were assessed to have a relatively high score for cohesiveness and adhesiveness, which suggests that further investigation may be required to study their suitability in dysphagia management.

Other negative quality parameters, such as oxidation, which leads to rancidity, are important for the shelf life and should also be considered. Salmon is a fatty fish and prone to lipid oxidation, responsible for the development of a rancid odour and taste, thus negatively affecting shelf life. Before sealing, the oxygen in the packaging was therefore replaced by 100% N_2_ as a modified atmosphere to inhibit aerobic microbial growth and lipid oxidation [32,33,34]. The rancidity attribute was not included in this assessment as it was not deemed as a relevant attribute when developing the vocabulary for the trained panel. From this, it is advised that in future work with fatty fish products, an analysis of oxidation should be included.

### 3.2. Colour Measurement

This study intended to produce a salmon product with a natural colour corresponding to fresh salmon. The individual colour parameters were evaluated among the protein variants during an extended period at day 8, 15, 22, 29, 36 and 43 (Figure 2). During the storage period, a significantly higher value for lightness (L*, *p* = 0.003) was found on day 22 than on day 15, 36, and 43. Chroma showed only minor changes during storage, except on day 22, with significantly lower values (*p* < 0.001). The hue values were higher (*p* < 0.001) on day 29 than day 15 and 22). The light colour of the sample could be explained by the use of whipping cream, sodium caseinate, and WPC 80, which increased the light reflection from the sample. Sample S was perceived as lighter in colour (L*, *p* = 0.006) and had lower chroma values (*p* < 0.001) than sample SFP. There was no significant difference in the hue value samples (*p* = 0.059) which indicates that the addition of fish hydrolysate did not contribute to any additional reddish colour to the sample SFP. Both samples were visually perceived as orangish in colour, which could be attributed to the natural colour of salmon.

### 3.3. Texture

To examine the shape and texture under normal serving and eating conditions, firmness was measured at 60 ± 2 °C (Figure 3), which is close to the recommended serving temperature at an institution (hospital/nursing home) [35].

Compared to day 1, the firmness decreased significantly (*p* < 0.001) for both S and SFP throughout the 43-day storage period. SFP had lower firmness for all sampling days compared to S. These results align with the sensory evaluation of softness (Figure 1). In general, the food ingredients, together with the different production processes used, strongly change the food structure [36,37]. Thus, the choice of hydrolysate is decisive, as it alters textural, rheological, and sensory properties. These alterations originate from the production of the hydrolysate, i.e., the protein source, degree of hydrolysis, enzyme used, pH, and globular structure [26,27,38,39,40].

Restructured seafood products are made from minced or chopped muscle tissue. These products may include additional ingredients or additives, resulting in new items with altered appearances, textures, or both [41]. In the recent years, there has been a continuous development of new restructured products from under-valued marine species or by-products [42,43,44]. Various techniques are being used to reconstruct seafood products to replicate other highly valued products [45]. Restructuring fish muscle is essential to tackle the undeniable limitations in the supply of high-value seafood products [42] and the necessity to utilize existing resources in a sustainable manner [15]. The texture was modified by blending the salmon raw materials and adding ingredients and liquids. This reduced the texture strength and the total protein concentration. Hydrocolloids and proteins were therefore added to rebuild the texture and obtain a high total protein concentration.

### 3.4. The IDDSI Framework Fork Test

An international standardization to identify and classify dysphagia was carried out using the IDDSI framework [46]. These tests were performed using basic tools such as a syringe, spoon, chopsticks, fork and even fingers. The purpose of such tests is to provide a platform for institution kitchens, hospitals, catering personnel, etc. which do not have access to advanced texture measuring instruments. The fork test is applied to categorize the food in IDDSI levels 5–7 by applying pressure on the sample [10].

The observation showed that a higher pressure was needed to blanch the thumb nail white in S samples than the SFP samples. The blanching of thumbnail was more visible in both samples S and SFP on day 1 compared to day 29 (Figure 4). The observations from the fork test comply with results from instrumental measurement of texture, where S and SFP were significantly (*p* < 0.001) firmer on day 1 than after day 22, and throughout the storage period of 43 days. Several researchers have used the fork pressure test to validate samples of newly developed products for dysphagia patients [47,48,49]. However, the fork test is a subjective method, and different individuals can influence results, the applied pressure, and holding time.

An instrumental method for the IDDSI fork test has now been published [47]. An instrumental fork is attached to the texture analyses to mimic the manual fork pressure test and provide a more controlled pressure. Technological developments like this can simplify texture standardizations and make it easier to test the effects of mixing hydrolysates, dairy proteins, and other ingredients in various combinations.

A large increase in the number of elderly people is expected in the coming years and there is a need for more research on texture modified products intended both for age care facilities and for the grocery trade. A review by [50] describes trends and strategies in developing and processing texture modified food products. They highlight examples of companies in this sector and the notable differences across countries, indicating that these products may represent an emerging market.

### 3.5. Heat Treatment

Elderly people are more susceptible to food-borne illnesses due to their compromised immune systems and the gradual loss of physical barriers. They have higher incidences of infection from foodborne pathogens such as *Listeria*, *Salmonella* and *Staphylococcus* [51]. The packages of salmon samples were therefore heated in a convection oven to a core temperature of 95 °C for 15 min per sample, inactivating non-spore-forming microorganisms. People with dysphagia are susceptible to foodborne disease [52], and safe pasteurization, packaging and storage is therefore crucial. Figure 5 shows accumulated F_90_^10^ values for three temperature probes, giving a variance in the core pasteurization values of the packages due to position in the oven and temperature fluctuations. The measured F_90_^10^ values were calculated based on D and z values from psychotropic non-proteolytic *Clostridium botulinum* type E (D_90_ = 1.66, z = 10) [53]. The pasteurization values ranged from 126 to 157, which were much higher than the heat load (D_90_ = 10) normally used for safety for a 6-log reduction in non-proteolytic *Clostridium botulinum* type E, indicating no survival of non-proteolytic *Clostridium* [53].

Since this is a chilled fish product (≤+4 °C), only a few psychrotrophic spore-forming bacteria species could survive pasteurization and then grow, like *Bacillus* spp. (minimum growth at approximately +4 °C) and non-proteolytic *Clostridium* type E or B, which can also grow after such conditions (minimum growth at 3.3 °C) [54,55]. *Bacillus* species are regularly not associated with raw fish, but are often isolated from starch, spices, etc. present in the ingredients [56]. *B. cereus* has a higher heat tolerance, with D_95_-values in the range of 1.5–36.2 min and z = 6.7–10.1 °C [57]. Spores from *Bacillus* may therefore survive the given heat treatment used in our experiments [58], although they were not detected in our shelf-life study. As their minimal growth temperature is 4 °C, they cannot reach infective doses at about 10^5^ cfu/g [53] in 29 days. In addition, the normal recommendation for reheating before serving is a minimum of 75 °C at the core, followed by serving at an eating temperature of 60 ± 2 °C before consumption [59]. At these temperatures, used in a controlled home care system, any vegetative bacteria, e.g., those germinated, as well as growth from spores, will be eliminated.

### 3.6. Shelf-Life Study

The two salmon products were packaged in HDPE trays, placed in master bags, and flushed with 100% N_2_ to reduce oxygen, simulating vacuum or modified atmosphere packaging. The residual oxygen in the HDPE trays was measured regularly for 43 days of storage and remained at 0.6 ± 0.1% throughout the period. The low oxygen concentration inhibited the growth of aerobic bacteria. During the 43-day shelf-life test, with products stored at 4 °C, the products were analyzed on day 8, 15, 22, 26, 36, and 43 for total viable counts on PCA and for aerobic and anaerobic spore-forming bacteria on blood agar plates. The stored samples were not reheated before microbial samples were taken (a core temperature of 75 °C is recommended as safe for eating) to evaluate the microbial status at the end of the chilled shelf-life period. The growth observed on PCA and blood agar plates for salmon samples with fish protein hydrolysate (SFP) was below the detection level (<100 cfu/g) throughout the 43-day storage period. This was also found for salmon samples without hydrolysates (S), except at storage day 36 and day 43. On day 36, one sample showed a total viable count of log 3.40/g on PCA. Similarly, one parallel sample, analyzed on day 43, showed a total count of log 3.32/g on aerobically incubated blood agar. The shelf life of some sous vide pasteurized fish products is found to be 3–6 weeks, based on heat treatment and post-cooking storage [60]. The storage temperature is crucial for safety; special concern must be taken in the absence of additional hurdles, such as the maintenance of pH ≤ 5 or water activity ≤ 0.97. At a storage temperature ≤ 5 °C, the recommendation is a maximum of 10 days for pasteurized products [61]. The pasteurization used in this work and the microbiological analyses show that it is possible to obtain safe prefabricated products that can be stored in refrigerated conditions (≤4 °C) without oxygen for 29 days.

## 4. Conclusions

Hydrolysates from residual products in the food industry may constitute a significant sustainable protein supplement for food products in the future. It is therefore important to acquire knowledge about how hydrolysate affects the physiochemical and sensory properties of new restructured products. This work has shown that it is possible to achieve a defined texture-modified product suitable for aged care facilities and for the retail market in the minced & moist category (IDDSI level 5) with 18.1% total protein, of which 1.7% consists of fish hydrolysate. The QDA showed that both the presence and absence of fish protein hydrolysate did not lead to a significant difference in bitterness but produced significant differences in the sensory attributes of fish flavour, softness, and adhesiveness. The texture analysis showed that the salmon product with fish protein hydrolysate, measured at 60 ± 2 °C, was significantly softer than those without. The product had good storage properties for possible chilled distribution. On day 29, the products were softer than on day 1. However, small variations in firmness of the products were observed during the storage period. The IDDSI fork test revealed that the products without fish protein hydrolysate required higher pressure to make the thumb nail blanch compared to products with fish protein hydrolysate. The heat treatment used in the different processes gave a low microbial load below detection level during 29 days at 4 °C and showed that the product is safe during a chilled distribution.

## Figures and Tables

**Figure 1 foods-14-00162-f001:**
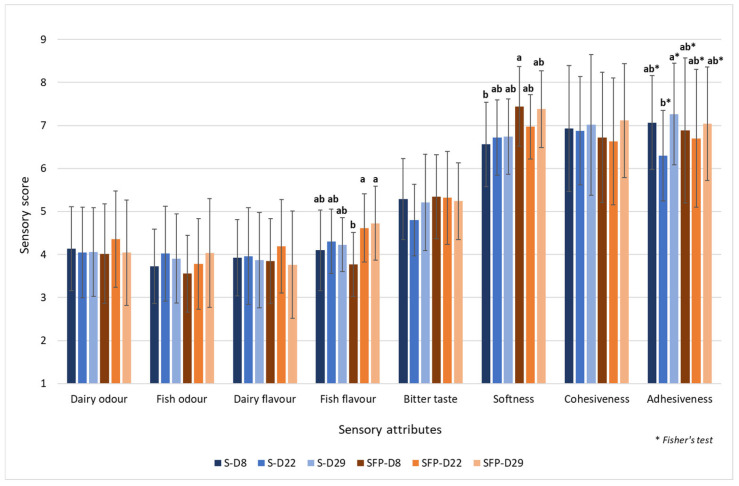
Sensory attributes evaluated by Quantitative Descriptive Analysis (QDA^®^) of texture-modified enriched salmon products, S (enriched with dairy proteins) and SFP (enriched with dairy proteins and fish protein hydrolysate) after storage for 8, 22, and 29 days at 4 °C. Sensory scores are presented as mean ± S.D. (*n* = 18). Means with different lowercase superscripts in the bar indicate significant differences calculated separately for each attribute. Attributes with no lowercase letters have non-significant differences.

**Figure 2 foods-14-00162-f002:**
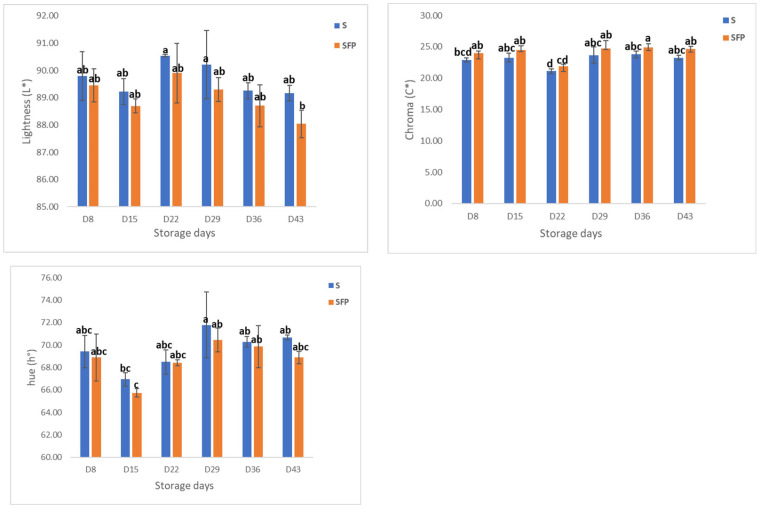
Changes in colour coordinates lightness L*, chroma C*, and hue angle h* of salmon sample S (with dairy proteins) and SFP (with dairy proteins and fish protein hydrolysate) for the 43-day storage period at 4 °C. The values are presented as a mean ± S.D. (n = 3). Means with different lowercase superscripts in the bar indicate significant differences.

**Figure 3 foods-14-00162-f003:**
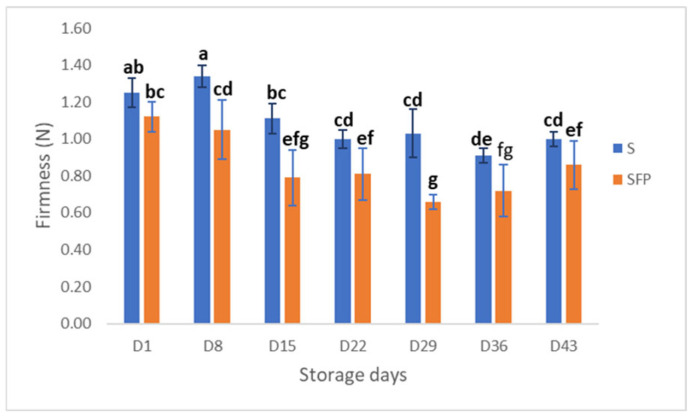
Firmness (N) of texture modified salmon products, sample S (with dairy proteins) and SFP (with dairy proteins and fish protein hydrolysate), for the 43-day storage period at 4 °C. The values are presented as a mean ± S.D. (*n* = 3 × 4). Means with different lowercase superscripts in the bar indicate significant differences.

**Figure 4 foods-14-00162-f004:**
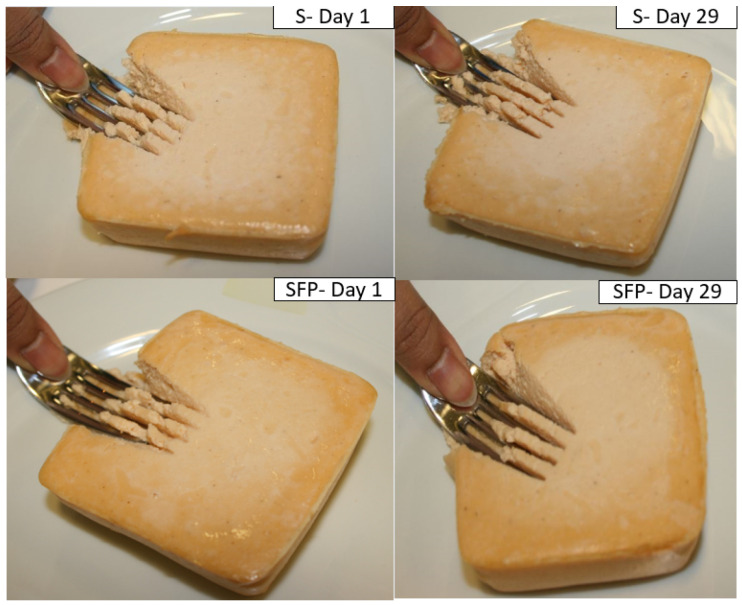
IDDSI fork test performed on salmon product S (with dairy proteins) and SFP (with dairy proteins and fish protein hydrolysate) on day 1 and day 29.

**Figure 5 foods-14-00162-f005:**
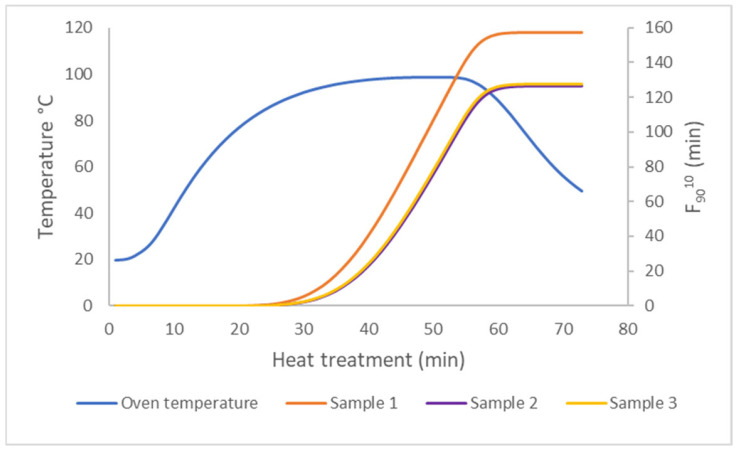
Temperature curve and F_90_^10^ values from three different samples with probes in the centre of the salmon product. The come-up time in the oven (blue line) was about 30 min, and cooling started at 55 min. Differences in the F-values are a result of different placements of the samples in the oven.

**Table 1 foods-14-00162-t001:** List of ingredients in texture-modified salmon products enriched with WPC 80 and sodium caseinate (sample S) and with WPC 80, sodium caseinate, and fish protein hydrolysate (sample SFP). Calculated total protein is included in the table. The values are given in percentages.

Ingredients	Recipe S (%)	Recipe SFP (%)
Precooked fish fillets	59.0	57.9
Salt	0.4	0.3
Whipping cream, 38% fat	23.6	23.2
Sunflower Oil	9.4	9.3
Cornstarch	0.6	0.6
Salmon fish protein hydrolysate (FPH)	-	1.7
Sodium caseinate	1.8	1.7
Whey protein concentrate 80	3.5	3.5
Spice mix ^a^	0.2	0.2
Fish stock powder	1.2	1.2
Locust bean gum	0.4	0.3
Total protein ^b^	17.8	18.1

^a^ Equal quantities (1:1:1:1) of ground fennel powder, ground ginger powder, mustard powder, and white pepper powder. ^b^ Norwegian Food Composition Database 2022. Norwegian Food Safety Authority. www.matvaretabellen.no, accessed on 24 September 2023.

**Table 2 foods-14-00162-t002:** Sequence of added ingredients, mixing speed, and time used during production of sample S (without FPH) and sample SFP (with FPH). The speed corresponds to between 1100 and 10,200 revolutions/min.

Addition Sequence	Ingredients	Mixing Speed	Mixing Time (min:s)
1	Precooked fish fillets	5	01:00
	Salt		
2	Whipping cream, 38% fat	10	02:30
	Sunflower Oil		
3	Cornstarch	10	00:15
4	Salmon fish protein hydrolysate (FPH) ^a^	10	00:10
5	Sodium caseinate	10	00:10
	Whey protein concentrate 80		
6	Spice mix ^b^	10	00:10
	Fish stock powder		
7	Locust bean gum	10	00:30

^a^ Used only in SFP recipes. ^b^ Equal quantities (1:1:1:1) of ground fennel powder, ground ginger powder, mustard powder, and white pepper powder.

## Data Availability

The original contributions presented in the study are included in the article, and further inquiries can be directed to the corresponding authors.

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
