# Peer review of "Fish Protein Hydrolysate as Protein Enrichment in Texture-Modified Salmon Products"

_foods, 2025, doi:10.3390/foods14020162_

Round 1
Reviewer 1 Report
Comments and Suggestions for Authors
see attachment

Author Response
1. The manuscript describes the study to develop a modified salmon product enriched with dairy and fish hydrolysate proteins formulated for patients with dysphagia. Technical, sensory and microbiological properties were characterized. The proposal and methodology are proper and consistent. The writing is high quality. I have no major revisions on this manuscript. Just a few minor revisions must be made.
Response: Thank you for the comments. We have accounted for the suggestions and edited the manuscript accordingly.
2. Please, add the conclusion in the abstract.
Response: Conclusions have been inserted into the manuscript and as a possible applied use of the results, one can now produce defined protein-enriched texture-adapted products for age care facilities and for commercial use.
3. Page 3. The uppercase letter “a” in Table 2 has no legend.
Response: This is now included in the manuscript.
4. Page 6. Fig.1. The significant differences indicated in the plot are between enriched salmon products and/or between storage time? Please, clarify this for all figures with statistical analysis.
Response: The differences are among the attributes between the samples. This clarification has been added to the manuscript.
Reviewer 2 Report
Comments and Suggestions for Authors
The paper is well written and the results are presented clearly, showing side-by-side comparison of the two formulations.
It would be helpful to include a short section on the translational potential of this applied research. Has this been applied in real world scenarios? If not, what is the roadmap moving forward?
Author Response
1. The paper is well written and the results are presented clearly, showing side-by-side comparison of the two formulations.
Response: Thank you for the review and the comment.
2. It would be helpful to include a short section on the translational potential of this applied research. Has this been applied in real world scenarios? If not, what is the roadmap moving forward?
Response: We have included a paragraph describing this topic in the end of Chapter 3.4, with a reference from 2024.
Reviewer 3 Report
Comments and Suggestions for Authors
In this study, restructured salmon products were developed with and without fish protein hydrolysate. Sensory and Colorimetric properties, firmness, IDDIS fork pressure, and Microbiological safety during shelf life were investigated. The results of the present work can advance current knowledge and be helpful for the industry. However, the scientific quality of this study is unsatisfactory. The research question on texture-modified products was not clear.
The objectives of this study are clearly stated, but the rationale is not clearly described. The methods in the text were reported in sufficient detail to allow for reproducibility. The data could support the interpretation of results and study conclusions. However, the data interpretation and analyses presented in the manuscript were insufficient. Only specific period data was shown and stated. The statistical reporting was not well-described in the text. Not all data was compared. Moreover, one-way ANOVA was not rational for some data comparisons in the text because there were two factors that affected the properties. The indication of the lowercase superscripts in the bar was not evident in the figure title.
The manuscript could benefit from adding the results of the microstructure and oxidation stability of the products.
In a word, the quality of this study is unsatisfactory, and the manuscript is not acceptable before major revision.
Additional comments were as follows:
(1) Line 160-161, what was the product temperature when the assessors evaluated the sensory properties? Did the product temperature affect the flavor compound concentration and texture of the products?
(2) Line 181-183, the product will not be consumed at that temperature. Decreasing the product temperature might result in the contract of the products. What was the firmness of the products when at the product serving temperature?
(3) Line 192, what was the product temperature?
(4) Line 208-211, what about the replicate times of the trials? Please make it clear.
(5) Line 260, it was not reasonable to consider in the future. The results of lipid oxidation were important in evaluating the quality of products.
(6) Line 264-265, why was the data on L, a,b not displayed in Figure 2?
(7) Line 292-294, The firmness of the products at 0 days was not shown in Figure 3.
(8) Line 295, what did ‘n=12’ mean? Twelve triplicates of fresh products were prepared on different days?
(9) Line 325-326, why was the fork test used only? How did the authors select the IDDIS level of the products?
(10) Line 335-341, the statement showed little relation with the topic.
